Hide and seek shark teeth in Random Forests: machine learning applied to Scyliorhinus canicula populations

http://orcid.org/0000-0003-0810-9783 Berio Fidji 1 2 fidji.berio@gmail.com
Bayle Yann 3
Baum Daniel 4
Goudemand Nicolas 1
Debiais-Thibaud Mélanie 2
1 Institut de Génomique Fonctionnelle de Lyon, École Normale Supérieure de Lyon , CNRS, UCBL, Lyon , France
2 Institut des Sciences de l’Évolution de Montpellier, CNRS, IRD, EPHE, Université de Montpellier , Montpellier , France
3 Université de Bordeaux, Bordeaux INP, CNRS, LaBRI , Talence , France
4 Department of Visual and Data-Centric Computing, Zuse Institute Berlin , Berlin , Germany
Quimbayo Juan Pablo
Electronic publication date: 2022 Jul 4
Publication date: 2022
Volume: 10
Electronic Location ID: e13575
Received 2022 Jan 18; Accepted 2022 May 22
Copyright: © 2022 Berio et al.
Copyright year: 2022
Copyright holder: Berio et al.
License: This is an open access article distributed under the terms of the Creative Commons Attribution License, which permits unrestricted use, distribution, reproduction and adaptation in any medium and for any purpose provided that it is properly attributed. For attribution, the original author(s), title, publication source (PeerJ) and either DOI or URL of the article must be cited.
License URL: https://creativecommons.org/licenses/by/4.0/

Keywords: Machine learning, Geometric morphometrics, Tooth morphology, Scyliorhinus canicula, Random Forests, Linear discriminant analysis, Sharks

Funding: Attractivité Nouveaux professeurs ENS de Lyon This work was supported by the ENS de Lyon “Attractivité Nouveaux professeurs” fund. The funders had no role in study design, data collection and analysis, decision to publish, or preparation of the manuscript.

==============================
Shark populations that are distributed alongside a latitudinal gradient often display body size differences at sexual maturity and vicariance patterns related to their number of tooth files. Previous works have demonstrated that Scyliorhinus canicula populations differ between the northeastern Atlantic Ocean and the Mediterranean Sea based on biological features and genetic analysis. In this study, we sample more than 3,000 teeth from 56 S. canicula specimens caught incidentally off Roscoff and Banyuls-sur-Mer. We investigate population differences based on tooth shape and form by using two approaches. Classification results show that the classical geometric morphometric framework is outperformed by an original Random Forests-based framework. Visually, both S. canicula populations share similar ontogenetic trends and timing of gynandric heterodonty emergence but the Atlantic population has bigger, blunter teeth, and less numerous accessory cusps than the Mediterranean population. According to the models, the populations are best differentiated based on their lateral tooth edges, which bear accessory cusps, and the tooth centroid sizes significantly improve classification performances. The differences observed are discussed in light of dietary and behavioural habits of the populations considered. The method proposed in this study could be further adapted to complement DNA analyses to identify shark species or populations based on tooth morphologies. This process would be of particular interest for fisheries management and identification of shark fossils.

Introduction

The recognition of disjunct shark populations has opened to new questionings on marine ecosystem connectivity and consequences of gene flows on species evolution. Shark population distributions are structured by ecological habits of species (e.g., degree of habitat fidelity), reproductive strategies (e.g., use of nursery areas by females), dispersal ability (e.g., type of reproduction, migratory behaviour), and environmental barriers to gene flow (e.g., oceanic basins, geological climatic events) (Lucifora et al., 2003; Rodríguez-Cabello et al., 2004; Portnoy et al., 2010; Veríssimo, McDowell & Graves, 2010; Karl, Castro & Garla, 2012; Kousteni et al., 2015). Populations of a given shark species sometimes display vicariance patterns in number of vertebrae (Gruber & Compagno, 1981) and tooth files (McEachran & Martin, 1977; Templeman, 1984; Lucifora et al., 2003). In some species, the total length of a specimen at sexual maturity also differs between populations, as in bonnethead sharks Sphyrna tiburo (Parsons, 1993; Lombardi-Carlson et al., 2003), shortspine spurdogs Squalus mitsukurii (Taniuchi & Tachikawa, 1997), starspotted smooth-hounds Mustelus manazo (Yamaguchi, Taniuchi & Shimizu, 1998, 2000), and cloudy catsharks Scyliorhinus torazame (Horie & Tanaka, 2002). Such differences in specimen size at sexual maturity that are reported among shark populations have been hypothesized to result from genetic or environmental constraints, or both, but the combination between these factors is diﬃcult to evaluate (Lombardi-Carlson et al., 2003).

When observed, these size differences at sexual maturity are often distributed alongside a latitudinal gradient and shark populations inhabiting higher and colder latitudes are significantly bigger (Leloup & Olivereau, 1951; Parsons, 1993; Taniuchi & Tachikawa, 1997; Yamaguchi, Taniuchi & Shimizu, 2000; Horie & Tanaka, 2002; Lombardi-Carlson et al., 2003; Capapé et al., 2014; Kousteni & Megalofonou, 2019). The warmer temperatures at lower latitudes (Blackburn, Gaston & Loder, 2008) are thought to limit the energy allowed for somatic growth by inducing increased energy expenditure (Parsons, 1993; Carlson & Parsons, 1997) and also trigger early sexual maturity (Parsons, 1993; Yamaguchi, Taniuchi & Shimizu, 2000; Goren, 2014).

Scyliorhinus canicula is an abundant benthic species in the eastern Atlantic Ocean (from Senegal to the UK) and Mediterranean Sea that inhabits depths from a few meters to 500 m (most commonly found around 110 m) (Compagno, 1984; Ellis & Shackley, 1997; Rodríguez-Cabello et al., 2004). Support for population differentiation was raised by morphometric and genetic diversity analyses within the distribution range of this species (Barbieri et al., 2014; Capapé et al., 2014; Kousteni et al., 2015). Population genetic structures have been attributed to the philopatric behaviour of S. canicula and to its low dispersal ability across basins (Leloup & Olivereau, 1951; Mellinger, Wrisez & Alluchon-Gérard, 1984; Rodríguez-Cabello et al., 2004; Barbieri et al., 2014; Capapé et al., 2014; Kousteni et al., 2015; Anastasopoulou et al., 2016; Kousteni & Megalofonou, 2019).

Despite genetic structuration, Mediterranean S. canicula populations exhibit very slight body size differences at sexual maturity (Barbieri et al., 2014; Capapé et al., 2014; Kousteni et al., 2015; Kousteni & Megalofonou, 2019). Conversely, populations differ greatly in body size at sexual maturity between the North Atlantic Ocean (Bristol Channel, UK) and the Mediterranean Sea (Leloup & Olivereau, 1951; Mellinger, Wrisez & Alluchon-Gérard, 1984; Rodríguez-Cabello et al., 1998; Kousteni, Kontopoulou & Megalofonou, 2010; Capapé et al., 2014; Kousteni & Megalofonou, 2019). Furthermore, there is currently no recognition of unequivocal morphological differences between S. canicula or any shark populations, other than body length at sexual maturity which differs along vast latitudinal gradient and separated marine environments. This is an important gap in our ability to follow population dynamics, notably for sharks that have long generation time and struggle withstanding or recovering from accidental catches and fishing (Smith, Au & Show, 1998; Cortés, 2000; Dulvy et al., 2014, 2021). Commercial frauds are regularly checked, but the identification of species and populations mainly relies on a framework based on DNA barcoding (Barbuto et al., 2010; Melo Palmeira et al., 2013; Almerón-Souza et al., 2018), whose cost, time, and dependence to a wet lab are limitations to the current inspection process. Therefore, we investigated the possibility to identify differences in tooth morphology between catshark populations, considering that shark tooth shape undergoes morphological changes with ontogeny, sexual maturation, and diet (Powter, Gladstone & Platell, 2010; Moyer & Bemis, 2016; Tomita et al., 2017; Cullen & Marshall, 2019; Berio et al., 2020) and that life history traits differ between Atlantic and Mediterranean S. canicula populations (Leloup & Olivereau, 1951; Ivory et al., 2004; Bendiab, Mouffok & Boutiba, 2012; Capapé et al., 2014).

This work tests a feature-based framework to reliably discriminate shark populations based on their tooth morphological characteristics. The assessment of shark tooth form differences is successfully achieved with geometric morphometrics (Whitenack & Gottfried, 2010; Soda, Slice & Naylor, 2017; Cullen & Marshall, 2019; Berio et al., 2020). Linear Discriminant Analysis (LDA) is a frequently used machine learning algorithm to discriminate between groups based on geometric morphometric data (Mitteroecker & Bookstein, 2011; MacLeod, 2017; Doyle, Gammell & Nash, 2018). The performances of LDA depend on a much higher number of items (e.g., teeth) as compared to the number of features (e.g., aligned coordinates). Yet, these conditions are diﬃcult to meet in biological datasets, which often implies a feature reduction step through a Principal Component Analysis (PCA) (Fort & Lambert-Lacroix, 2005; Pechenizkiy, Puuronen & Tsymbal, 2006; Sheets et al., 2006; Archer & Kimes, 2008). Moreover, subtle discriminant patterns can also be missed when reducing data dimensionality to the first PCA axes prior to a classification task (MacLeod, 2018). Traditional and geometric morphometric studies use several machine learning algorithms for classification, but often focus on the classification performances–rather than on interpretable features (Santos, Guyomarc’h & Bruzek, 2014; Navega et al., 2015; Doyle, Gammell & Nash, 2018; Courtenay et al., 2019). As opposed to LDA, Random Forests (Breiman, 2001) usually outperform other machine learning algorithms for supervised classification (Caruana & Niculescu-Mizil, 2006; Domínguez-Rodrigo & Baquedano, 2018; Doyle, Gammell & Nash, 2018; Püschel et al., 2018; Courtenay et al., 2019). Furthermore, Random Forests provide intuitive and interpretable importance measures of feature contribution to classification, is highly resistant to overfitting, and do not require feature reduction prior to the analysis to achieve good performances (Díaz-Uriarte & Alvarez de Andrés, 2006; Archer & Kimes, 2008; Doyle, Gammell & Nash, 2018).

This work is a proof of concept that shark populations can be discriminated based on tooth morphology. We take advantage of the geometric morphometric and machine learning methods to test for differences between S. canicula teeth from northeast Atlantic and Mediterranean populations that exhibit clear body size differences at sexual maturity (Capapé et al., 2014; Kousteni & Megalofonou, 2019). By using Random Forests to classify the teeth of S. canicula population samples of different geographic origin, we aim not only to challenge the traditional geometric morphometrics workflow, but also to provide indications of discriminant tooth form features between S. canicula populations. Our work aims to provide an alternative tool for discrimination of shark species and populations based on dental features and to set the basis for future research.

Materials and Methods

Sampling

S. canicula populations were sampled in 2018 and 2019 in two localities separated by over 2,000 nautical miles: off Roscoff (France, northeast Atlantic Ocean) and Banyuls-sur-Mer (France, western Mediterranean Sea). The Atlantic specimens were sampled at the Roscoff fishmarket and were provided by the Station Biologique de Roscoff and the University of Montpellier. The Mediterranean specimens were incidentally caught during experimental surveys of the Observatoire Océanologique de Banyuls-sur-Mer or were formerly euthanized for independent experiments led by the Observatoire Océanologique de Banyuls-sur-Mer and the University of Montpellier. Biological samples were preserved in 70% ethanol. Specimens were selected based on their total body length (TL, from the tip of the snout to the tip of the tail) to account for three ontogenetic stages within each population, hereafter referred to as “hatchling”, “juvenile”, and “mature”. The TL of mature Atlantic and Mediterranean specimens was selected according to Ivory et al. (2004) (>53.5 cm TL for males, >57 cm TL for females; reference lengths for 50% maturity) and Leloup & Olivereau (1951) ( ≥40 cm TL for both sexes; reference length for “frequently” observed maturity), respectively. Hatchling specimens were euthanized just after hatching. Mediterranean juveniles were selected between hatchling [>9 cm TL for both sexes (Leloup & Olivereau, 1951)] and sexually mature [<37.5 cm TL for both sexes; reference length for first sexual maturity (Leloup & Olivereau, 1951)] stages. Atlantic juveniles were selected between hatching [>10.5 cm TL for both sexes (Ellis & Shackley, 1997)] and first sexual maturity [<49 cm and <52 cm in males and females, respectively (Ellis & Shackley, 1997; Ivory et al., 2004)]. The Mediterranean sample is composed of six hatchling (three females, three males; 9.1 ± 0.3 cm TL; 8.8 to 9.5 cm TL), 10 juvenile (five females, five males; 26.8 ± 4.1 cm TL; 21 to 31 cm TL), and nine mature (five females, four males, 42.9 ± 2.7 cm TL; 40 to 47 cm TL) specimens. The Atlantic sample includes 11 hatchling (six females, five males, 11.9 ± 1.4 cm TL; 10.2 to 13.9 cm TL), 10 juvenile (five females, five males, 34.4 ± 1.5 cm TL; 32 to 36 cm TL), and 10 mature (five females, five males, 58.9 ± 2.7 cm TL; 56 to 64 cm TL) specimens. The maturity assessment of specimens was not conducted because only the heads were collected for the juvenile and sexually mature specimens after body length was recorded in marine stations. Attempts were made to equally sample specimens from both sexes (F, female; M, male) within each category. In addition, we estimated the approximate age of the specimens using the von Bertalanffy growth parameters for the Atlantic and Mediterranean populations provided by Ivory et al. (2004) and Bendiab, Mouffok & Boutiba (2012) respectively because these studies provide sex-specific growth curves. Ivory et al. (2004) estimated von Bertalanffy parameters based on vertebral growth increment counts, whereas Bendiab, Mouffok & Boutiba (2012) estimated these parameters by analyzing length-frequency distributions.

We used the growth parameters ( K, coeﬃcient of growth; L∞, asymptotic length; Lt, length at age t; t0, theoretical age at which the size is zero) from literature data (Ivory et al., 2004; Bendiab, Mouffok & Boutiba, 2012) (Fig. 1) to compute von Bertalanffy growth curves with the following growth equation: Lt=L∞[1−e−K(t−t0)].

Figure 1 Growth curves of Scyliorhinus canicula from northeast Atlantic Ocean and Mediterranean Sea.

Von Bertalanffy growth parameters retrieved from [1] Ivory et al. (2004) and [2] Bendiab, Mouffok & Boutiba (2012). Markers represent the specimens used in the current study. Black, Mediterranean specimens; grey, Atlantic specimens. H, hatchling; J, juvenile; M, mature.

The ages were estimated for each specimen according to its sex and population and the age estimates were subsequently averaged per sex, ontogenetic stage, and population. We estimated the age of Mediterranean hatchlings to be 0 year. Female and male Atlantic hatchlings are 0.2 and 0.1 year old, respectively. Juvenile females in the Mediterranean population are 0.8 year old and juvenile males are estimated to be 1.1 year old. Mean age of juveniles from the Atlantic is 3.1 years for both sexes. Within the Mediterranean population, sexually mature females and males are 3.7 and 3.9 years old, respectively. The sexually mature specimens from the Atlantic were estimated to be 9.9 years old for females and 7.9 years old for males. However, the parameters for the Atlantic and Mediterranean specimens were estimated with different methods, the results of which depend either on calcification degree (Natanson et al., 2018) and periodicity, or on sample size and reproducibility of the results (Schwamborn, Mildenberger & Taylor, 2019). The estimated ages of the specimens sampled in this study are therefore not strictly comparable and should be interpreted in light of the method limitations, but are useful indicators of differences between the populations and ontogenetic stages considered.

Data acquisition

The jaws were microCT scanned using a Phoenix Nanotom S or an EasyTom 150 with voxel sizes ranging from 6.0 to 30.8 µm and 3D volumes were reconstructed using the phoenix datos x2 (v2.3.0) reconstruction or xact softwares (v11025). The surfaces are available in Berio et al. (2022). 3D surfaces of right palatoquadrate (upper) and Meckelian (lower) teeth were generated with the Amira software (v6.5), extracted using the ContourTreeSegmentation module from the AmiraZIBEdition software (v2018.28) (Stalling, Westerhoff & Hege, 2005), and were labelled according to their mesio-distal position along the jaw ( NAtlantic = 1,757 and NMediterranean = 1,542, Fig. 2A). Seven landmarks (respectively numbered 1, 13, 15, 17, 19, 21, and 33 in Fig. 2, Supplemental Materials 1 and 2) and 31 semilandmarks were placed on each 3D tooth surface. The semilandmarks were made denser on the lateral sides of the teeth, where modifications of accessory cusp number are reported during S. canicula ontogeny (Debiais-Thibaud et al., 2015) (Fig. 2B and see Berio et al. (2020) for similar trend in S. stellaris).

Figure 2 Labelling and landmarking Scyliorhinus canicula teeth.

(A) Mesio-distal numbering of right palatoquadrate and Meckelian teeth in a mature male from the North Atlantic population (59 cm TL); (B) Right Meckelian lateral tooth of the specimen in A with numbering of landmarks (red dots) and semilandmarks (small circles on grey lines). Dorsal (left) and mesial (right) views. A, anterior; D, distal; L, left; Lab., labial; Ling., lingual; P, posterior; R, right.

Data analyses

A Generalised Procrustes Analysis was performed on 3D coordinates that were formerly preprocessed following Berio & Bayle (2020). Semilandmarks were allowed to slide based on minimised bending energy (Bookstein, 1991). The structure of the dataset was first investigated through a PCA and centroid size patterns, and tooth centroid sizes were used as a proxy for tooth size. The centroid size is computed as the square root of the sum of squared distances between all landmarks and semilandmarks and the centroid of a form (Webster & Sheets, 2010; Klingenberg, 2016). The slopes of allometric patterns between populations of same sex was assessed with ANCOVAs and the interaction between shape data and size was tested using linear regression models. The shape data used to test for allometry were PC axes, whose number was determined following the procedure described by Evin et al. (2013). The relationship between shape data and size was tested using centroid size values of teeth and the specimens TL. Allometric vectors were computed with linear models and pairwise comparisons allowed to compare lengths and angles between the vectors of populations of same sex.

Supervised classification of teeth from Atlantic and Mediterranean populations was first achieved with LDA on the same PC axes used for testing allometry, without and with tooth centroid sizes [respectively tooth shapes and forms (Klingenberg, 2016)]. The results from the LDA model were compared with those obtained with Random Forests on tooth shape (Procrustes coordinates) and form (Procrustes coordinates + tooth centroid sizes) (Breiman, 2001). A five-fold cross-validation was performed in both methods (train set = 80% and test set = 20%).

In Random Forest models, the number of trees was set to 500 and the minimal node size was set to one. The models were allowed to sample among 114 (shape) to 115 (form) variables to split each internal node. A good fit of the models was determined through the comparison between the accuracies reached on the train and test sets, and for both models (on shape and form), the difference between these metric values was 0.2%. The feature importance for the classification was assessed with a measure of Mean Decrease Accuracy (MDA) and features with importance values ≥1.0% were commented (Breiman, 2001).

The classification performances reached by the models were compared based on three metrics: the accuracy, precision, and recall. The accuracy is an average of precision values for the Atlantic and Mediterranean populations. The precision is the number of items correctly assigned to a group (e.g., Atlantic), as compared to all items (Atlantic and Mediterranean) classified in this same group (Atlantic). The recall is interpreted as the number of specimens correctly assigned to a group (e.g., Atlantic), as compared to the total number of specimens actually belonging to this class (Atlantic). Additionally, a detailed confusion matrix is proposed to identify which teeth of males, females and hatchling, juvenile, and mature specimens were classified the best.

The classification results on shape and form were compared to evaluate the contribution of the centroid size to population discrimination. The overall results of LDA and Random Forest models on tooth shape and form were further compared.

The geometric morphometric analyses were performed using the geomorph package (v3.1.1) and supervised classification was computed using the MASS package (v7.3.53) for LDA and the randomForest package (v4.6.14) for Random Forests with R software (v4.0.3) (Liaw & Wiener, 2002; Venables & Ripley, 2002; Adams, Collyer & Kaliontzopoulou, 2019; R Core Team, 2020).

Results

Visual description

The tooth diversity presented in Fig. 3 is a selection of examples among the whole variation observed in the dataset. It provides a broad overview of the main morphological differences and associated factors in S. canicula tooth forms. Differences in mesio-distal location of a tooth are usually linked with an addition of accessory cusps (Figs. 3A and 3B) and an increase of the main cusp bending. Gynandric heterodonty at sexually mature stage is characterised by more accessory cusps in females as compared to males (Figs. 3C and 3D). Along the ontogeny, new teeth undergo size increase, as well as the addition of accessory cusps, except after sexual maturation (Figs. 3E and 3F), when males experience a decrease in number of accessory cusps, as opposed to females. The interpopulational tooth form differences involve less accessory cusps and blunter teeth in the Atlantic population than in the Mediterranean population (Figs. 3G and 3H).

Figure 3 Examples of tooth morphological differences in Scyliorhinus canicula.

(A and B) Form differences between tooth files: Meckelian teeth from file 5 (A) and 20 (B) of a North Atlantic mature female; (C and D) Form differences between sexes: palatoquadrate teeth from file 15 of Mediterranean mature male (C) and female (D); (E and F) Form differences between ontogenetic stages: Meckelian teeth from file 5 of North Atlantic hatchling (E) and juvenile (F) males; (G and H) Form differences between populations: Meckelian teeth from file 15 of North Atlantic (G) and Mediterranean (H) juvenile males. Scale bars are 100 μm.

We report significantly more Meckelian tooth files in Atlantic juvenile females ( 23±2) as compared to the Mediterranean ones ( 20±1) (permutation t test, t = 3.29, p-val = 0.04), all other interpopulation tests being not statistically significant (permutation t tests, p-vals >0.11).

Geometric morphometrics

The first two principal components gather 61% of the total variation in the dataset. The Atlantic and Mediterranean populations are not visually discriminated in the morphospace (Fig. 4). Extreme shapes for PC1 and PC2 suggest that most variation of the dataset relates to the number of accessory cusps, to their relative size compared to the main cusp, and to the mesio-distal bending of the main cusp (Fig. 4). In addition, shape variation along PC1 might also involve the relative width of the crown base, as compared to the main cusp height (Fig. 4).

Figure 4 PCA with all Scyliorhinus canicula teeth contained in the dataset.

Black, teeth from Mediterranean specimens; grey, teeth from North Atlantic specimens. Wireframes are the extreme shapes for PC1 and PC2.

The tooth centroid size patterns in the S. canicula dataset are exemplified for male palatoquadrate teeth (Fig. 5), but similar trends are observed for both jaws and sexes. In hatchlings, the tooth centroid size patterns display no variation along the jaw nor visual differences between populations (Fig. 5). The patterns overlap between juveniles of both populations, while tooth centroid sizes of mature Atlantic males are 35% bigger than those of mature Mediterranean males (Fig. 5). In mature specimens of both populations, the tooth centroid sizes are overall higher for the Atlantic population, whose body length is also higher as compared to Mediterranean specimens (Figs. 1 and 5). However, no such trend is observed for juveniles, whose differences in body length are not reflected by their tooth centroid size patterns (Figs. 1 and 5).

Figure 5 Tooth centroid size patterns of North Atlantic and Mediterranean populations of Scyliorhinus canicula: Example of male palatoquadrate teeth.

Black, teeth from Mediterranean specimens; grey, teeth from North Atlantic specimens. The centroid size is in mm.

The first 12 PCs (93.43% of the total variation) are selected to represent the tooth shape in the following statistical tests and are used as LDA features. The tooth centroid size and the TL of specimens significantly impact the tooth shape in all subgroups (e.g., Mediterranean × females) (One-Way MANOVAs, p-vals <2.20e−16), meaning for example that bigger teeth in larger specimens constrain the global shape of their teeth. The slopes of allometric patterns significantly differ between females and between males of both populations, indicating that the relationship between tooth size and shape is not equivalent between populations (Two-Way ANCOVAs, p-vals <2.20e−16). The strength of relationship between shape and tooth centroid size is similar between females from Atlantic and Mediterranean populations (linear regressions, adjusted R-squared respectively of 0.83 and 0.81, p-vals <2.20e−16), as well as between Atlantic and Mediterranean males (linear regressions, adjusted R-squared respectively of 0.86 and 0.84, p-vals <2.20e−16). The allometric relationship between shape and TL is also similar between Atlantic and Mediterranean populations (linear regressions, adjusted R-squared respectively of 0.73 and 0.69 in females and of 0.82 and 0.81 in males, p-vals <2.20e−16), which indicates that the tooth shapes are similarly modified over the ontogeny in both populations.

Supervised classification

General performances

The LDA classification reached an accuracy of 64.5±0.7% with tooth shapes and 74.6±1.2% with tooth forms. The precision is similar for both populations, whereas the recall is much higher for the Atlantic population as compared to the Mediterranean one (Fig. 6). Better performances are also achieved on tooth forms as compared to tooth shapes (Fig. 6).

Figure 6 Classification performances of LDA and RF algorithms on Scyliorhinus canicula teeth from a North Atlantic and a Mediterranean population.

(A) Precision values; (B) Recall values. Black, teeth from Mediterranean specimens; grey, teeth from North Atlantic specimens. LDA, Linear Discriminant Analysis; RF, Random Forests.

The classification task performed by the Random Forests reaches an accuracy of 81.7±1.7% with tooth shapes and 86.9±1.4% with tooth forms, indicating a significant contribution of the centroid size information at improving the discrimination between populations. This means our protocol allows differentiating Mediterranean from Atlantic S. canicula tooth forms with 85.5% to 88.3% accuracy. The precision values are similar between both populations, but the recall values are better for the Atlantic population than for the Mediterranean one (Fig. 6).

Subclass results

To get further indications of the classification performances, we detail the confusion matrix for each sex-stage subclass in the dataset (Table 1). We remind, however, that these values are still computed based on the population class (Atlantic and Mediterranean) only and that the performances for subclasses (e.g., Atlantic female hatchling) are detailed after the classification process.

Table 1 LDA and RF performances of tooth shape and form classification from North Atlantic and Mediterranean Scyliorhinus canicula populations.

Atl., Atlantic; Med., Mediterranean.

	LDA	Random forests	
	Shape	Form	Shape	Form	
	Precision (%)	Recall (%)	Precision (%)	Recall (%)	Precision (%)	Recall (%)	Precision (%)	Recall (%)	
Atl. (female hatchling)	57.1	63.8	63.5	68.1	64.4	80.6	82.8	80.0	
Atl. (female juvenile)	74.5	75.0	88.5	69.4	81.4	81.4	88.5	83.1	
Atl. (female mature)	73.6	61.3	78.8	84.0	81.5	84.3	85.7	84.6	
Atl. (male hatchling)	49.3	59.6	60.7	59.6	70.4	79.2	83.3	83.3	
Atl. (male juvenile)	78.4	79.0	86.1	73.4	94.3	97.1	91.9	89.5	
Atl. (male mature)	50.9	92.2	65.2	94.8	77.9	92.3	84.9	100	
Med. (female hatchling)	41.9	35.3	52.2	47.1	69.6	50.0	66.7	70.6	
Med. (female juvenile)	65.7	65.1	67.9	87.7	76.8	76.8	79.2	85.7	
Med. (female mature)	72.8	82.4	86.5	81.9	83.1	80.2	84.6	85.7	
Med. (male hatchling)	45.2	35.2	58.2	59.3	80.0	71.4	72.2	72.2	
Med. (male juvenile)	71.7	71.0	70.3	84.1	95.7	91.7	87.7	90.5	
Med. (male mature)	57.1	10.5	90.4	49.0	88.9	70.2	100	75.8	

With LDA, the lesser precision values are achieved for hatchlings and mature males with shape data and for hatchlings only with form data (Table 1). The less complex dental morphologies of the dataset are also visually identified in these groups that display one or three tooth cusps. Recall values are low for Mediterranean hatchlings and mature males for which the recall reaches 10.5% with shape data (Table 1). This means that amongst all Mediterranean mature males only 10.5% are detected as such by the model.

The Random Forests models on shape and form data achieve better classification performances for juvenile and mature specimens than for hatchling ones (Table 1). The model with form data also reaches 100% precision for Mediterranean mature males and 100% recall for Atlantic mature males (Table 1).

In most subclasses, better performances are obtained with form data as compared with shape data (Table 1). In some cases, however, the classification of form data confuses the models, which reach identical or lesser performance values than with shape data. The tooth centroid size for example does not improve the recall of Atlantic hatchling males after a LDA (Table 1). Lesser performances with form rather than shape data are also reported with LDA performed on teeth of Atlantic juvenile males and females (recall values) and with Random Forests performed on teeth of Atlantic and Mediterranean juvenile males (precision and recall values) (Table 1).

With shape data, the most important feature lies in semilandmark 11 (4.5% in x), followed by semilandmark 24 (1.5% in x and 1.4 in y) and landmarks 1, 13, and 19 (1.5% in y, y, and x respectively) (Supplemental Material 1). Semilandmarks 12, 22, and 23 also account for more than 1.0% in accuracy (1.0% in x and 1.3% in y, 1.0% in y, and 1.1% in y, respectively) (Supplemental Material 1). With form data, however, the feature contributing the most to the classification is centroid size (12.3%) (Supplemental Material 2). The following most important features with form data are semilandmark 11 (5.7% in x and 1.6% in y), semilandmark 24 (2.1% in x), landmark 19 (1.1% in x and 1.8% in y), semilandmark 23 (1.7% in x), semilandmark 12 (1.2% in y), landmarks 1 and 13 (1.0% in y), and semilandmark 38 (1.0% in z) (Supplemental Material 2).

Discussion

Visual descriptions and GM struggle to discriminate between populations

This work highlights that inter-population tooth differences lie in the lateral cusps. We first visually examined S. canicula teeth from both populations and, except in hatchlings, the teeth in the Mediterranean population appear sharper than in the Atlantic at all locations and usually display more accessory cusps for equivalent mesio-distal positions along the jaw.

On centroid size patterns, very few elements discriminate between the two populations: hatchling and juvenile specimens of both populations display similar tooth centroid size patterns and amplitude along the jaw. The main difference between populations arises between juvenile and mature stages because mature Atlantic specimens have teeth whose centroid size is about 35% higher than those of mature Mediterranean S. canicula. The similarities between the tooth centroid size patterns of Atlantic and Mediterranean hatchlings are consistent with their very close estimated age and TL (Fig. 1). In mature specimens also, the amplitude delta in tooth centroid size patterns can be easily interpreted in light of their TL and age differences (Fig. 1). Yet, the age estimations we provide remain approximative because we chose among several studies on S. canicula growth parameters. The similar amplitude of tooth centroid size patterns between juveniles of both populations could also be a consequence of the very close ranges of size of our specimens (Fig. 1). Regarding allometric patterns, differences in slopes were detected between populations, meaning that the difference of increase between tooth centroid size and body size is not the same between the Atlantic and Mediterranean populations considered.

In summary, the reported differences between the populations’ tooth centroid size patterns seem to be related to the TL of mature specimens, but they do not discriminate between both populations among hatchlings and juveniles.

The contribution of Random Forests to decipher inter-population differences

Machine learning models have already improved the understanding of subtle structures in geometric morphometric data (Lorenz, Ferraudo & Suesdek, 2015; Soda, Slice & Naylor, 2017; Doyle, Gammell & Nash, 2018; Courtenay et al., 2019; Quenu et al., 2020; Barone et al., 2021), especially when the shapes between groups share common quadrants in a morphospace, as it is the case for the S. canicula populations considered in this study. However, even though the algorithms perform well at classifying geometric morphometric data, the choices made by these models to make groups are usually unknown and deprived of biological meaning (Lorenz, Ferraudo & Suesdek, 2015; Quenu et al., 2020).

In traditional geometric morphometrics, supervised classification is performed with an LDA on several PC axes based on raw shape data. We aimed to compare the results from this traditional workflow with the classification performances obtained with Random Forests on raw shape data. Doyle, Gammell & Nash (2018) already compared the classification performances of LDA and Random Forests on the shells of populations of common periwinkles (Littorina littorea) distributed in different niches. They recommended the use of Random Forests over LDA because the former is more straightforward and robust, does not make assumptions about the data nor necessitate the user to check the violation of LDA assumptions (Doyle, Gammell & Nash, 2018). However, the models of Doyle, Gammell & Nash (2018) do overfit and the results obtained might not be optimal.

Overall, LDA models achieve lower performances than Random Forests with our dataset, which can be due to information reduction. We used LDA for classification as in the majority of geometric morphometric articles, e.g., without considering the data distribution nor the homogeneity of variance. That such criteria for optimality are not met does not prevent from performing an LDA. The LDA algorithm is robust to such violations and still achieves good performances when assumptions of normality and common covariance matrix among groups are not met (Lachenbruch & Goldstein, 1979; Li, Zhu & Ogihara, 2006). Furthermore, Doyle, Gammell & Nash (2018) showed similar performances between LDA and Random Forests on geometric morphometric data and conclude that LDA is robust enough to the abovementioned violated conditions to achieve good classification performances.

The contribution of Random Forest models to our work is four-fold. It requires less preprocessing steps than LDA, performs better than LDA at the classification task, achieves good performances at classifying teeth from two S. canicula populations based on raw data, and determines the most discriminant features to perform this task. Among landmark and semilandmark features, it is clear that the most discriminant information is contained in two dimensions (x and y), whereas the third dimension brings less information. This implies that running the models with 2D landmark data would have probably achieved similar classification performances as those obtained here, as this has also been evidenced in some traditional geometric morphometric studies (Cardini, 2014; Buser, Sidlauskas & Summers, 2018; Wasiljew et al., 2020). Nevertheless, 3D landmarking avoids parallax biases (Cardini, 2014) and none of the landmarks and semilandmarks caused a decrease of the metrics, indicating that even though the z dimension contains little information, it still cannot be considered as noise. Furthermore, the landmarks (1, 13, 19) and semilandmarks (11, 12, 23, and 24) contributing the most to the classification are located at geometrical extrema, at extreme mesial and distal locations of the teeth and on the lateral edges, where accessory cusps emerge. It is also likely that the spacing of these points makes them useful to the algorithm and that spatially close points would bring less information. Overall, the results show that a few points, especially on the lateral sides of the teeth, provide enough information to represent most differences between the tooth shapes of the two S. canicula populations considered. This is consistent with our visual inspection on the variation of accessory cusps number.

The use of form features instead of shape greatly improves the overall classification. We expected such impact in the classification of mature specimens due to the amplitude differences of their centroid size patterns (likely caused by TL and putative age discrepancies, Fig. 1). However, the reason for the significant contribution of tooth centroid size to the classification improvement of the teeth of juveniles and hatchlings is less intuitive. We first assume that the very slight differences in tooth centroid size patterns we visually interpret as part of the inter-populational variability might actually be considered useful information for the model to discriminate between populations. The centroid size is theoretically independent of shape, however, the placement and density of landmarks and semilandmarks modify the contribution of certain parts of a tooth (e.g., the lateral sides of the crown as compared to the main cusp) to the centroid size value. Thus, we assume that slight changes in centroid size values between populations might also convey relevant shape information for the Random Forest model, which is cryptic to the observer. For a minority of subclasses, however, the addition of centroid size to shape features does not improve the classification performances or diminish them. In the first case, the centroid size is probably too similar between two subgroups (e.g., Atlantic and Mediterranean hatchling males) to allow the model to discriminate against the populations. In the second case, the centroid size information confuses the model, which might indicate that some form data contain more noise than in other groups, which could be overcome by increasing the sampling effort for the subclasses considered (e.g., Atlantic and Mediterranean juvenile males).

Ecological origins of anatomical divergence between populations

The inter- and intraspecific diversity of shark tooth shapes correlates with their feeding behaviour. Molariform teeth, for example, help crushing hard-bodied preys, while cutting teeth allow to remove pieces from larger items (Cappetta, 1986).

S. canicula is considered a generalist predator with opportunistic behaviour, whose favourite preys are teleosts, cephalopods, and crustaceans but it also occasionally feeds on macroalgae and echinoderms (Olaso, 1998; Mnasri et al., 2012; Kousteni, Karachle & Megalofonou, 2017; Kousteni et al., 2018). Within the same population, however, slight differences in diet composition can occur between specimens of different sex and ontogenetic stages (Lyle, 1983; Olaso, 1998; Rodríguez-Cabello, Sánchez & Olaso, 2007; Mnasri et al., 2012; Kousteni, Karachle & Megalofonou, 2017; Kousteni et al., 2018). S. canicula also displays seasonal diet shifts that differ according to the geographic area: more cephalopods are eaten in the winter than in autumn by a Cantabrian Sea (North Atlantic Ocean) population, while a population from eastern Mediterranean Sea feeds most on teleosts in spring and on molluscs in autumn (Olaso, 1998; Kousteni, Karachle & Megalofonou, 2017). Molluscs are far less important in the diet composition of Atlantic populations than in Mediterranean ones, whereas Atlantic specimens feed more on teleosts (Kousteni, Karachle & Megalofonou, 2017). However, the diet differences between several Atlantic and Mediterranean S. canicula populations are probably due to the variability of available prey items in contrasting habitats, as suggested by the opportunistic behaviour of this species (Lyle, 1983; Kousteni, Karachle & Megalofonou, 2017). Additionally, we show in this work that S. canicula tooth shapes differ between one Atlantic and one Mediterreanean population. If this variation is under functional selection, diet differences would correlate with distinct tooth forms: sharper teeth of the Mediterranean population may enhance grasping molluscs such as cephalopods, whereas teeth of Atlantic specimens might perform better at catching benthic teleosts. However, the specific diet of the specimens sampled in this study is not known and the relationship between the tooth morphologies depicted here and broad dietary trends in all Atlantic and Mediterranean S. canicula remains hypothetical. Overall, several studies already suggested that the diet composition of S. canicula, as well as of other elasmobranchs, is correlated to the body size (Lyle, 1983; Bethea et al., 2006; Ellis & Musick, 2007; Borrell et al., 2011; Šantić, Radja & Pallaoro, 2012; Kousteni, Karachle & Megalofonou, 2017). Furthermore, our results support the correlation between tooth shape and ontogeny, and thus body length in S. canicula, which could suggest that there is an association between diet shifts and tooth shape modifications over a specimen lifetime.

S. canicula has a philopatric behaviour and most specimens do not move further than 30 km over the years (Rodríguez-Cabello et al., 2004). Additionnally, shared haplotypes between specimens caught off Portugal and specimens from the Mediterranean Sea demonstrate past communication occurrences between the populations, probably resulting from colonisation events from the Atlantic or retention of ancestral polymorphism (Kousteni et al., 2015; Ramírez-Amaro et al., 2018). There is evidence of multiple genetic stocks of S. canicula within the Mediterranean Sea (Barbieri et al., 2014; Kousteni et al., 2015). Yet, molecular studies suggest that the Siculo-Tunisian Strait may allow gene flow between the eastern and western Mediterranean populations, whereas genetic exchanges are currently very low between the Atlantic and Mediterranean populations of S. canicula (Barbieri et al., 2014; Kousteni et al., 2015). The communication between these Atlantic and Mediterranean populations would indeed only be permitted through the Strait of Gibraltar, yet bottom topography may limit the migration between populations to a few specimens (Ramírez-Amaro et al., 2018). Therefore, the differences in tooth morphology between the Atlantic and Mediterranean population samples of S. canicula is probably related to the species genetic differentiation recorded between these locations (Barbieri et al., 2014). This assumption could be further tested by examining the differentiation at both molecular and tooth morphology level between these stocks.

Conclusions

We combined geometric morphometrics with a machine learning approach to discriminate between teeth of population samples of S. canicula from the northeastern Atlantic Ocean and the Mediterranean Sea. The traditional framework used in geometric morphometrics reached lesser performances at distinguishing the tooth shapes of S. canicula specimens from the two populations. Nevertheless, these shape data combined with centroid sizes allow a Random Forest model to classify S. canicula teeth with up to 100% precision. This framework should be further tested in more S. canicula populations to decipher tooth morphological differences between spatially and genetically close-related populations. We hope this emergent framework to be further tuned by ichthyologists by including geographical parameters and life history traits to discriminate between subtle tooth morphologies and to provide a basis for facilitating the identification procedures of fish stocks and improving fisheries management.

Supplemental Information

Supplemental Information 1 Mean decrease accuracy values with tooth shape data.

x, y, and z are the 3D dimensions. L, landmark; SL, semilandmark.

Click here for additional data file.

Supplemental Information 2 Mean decrease accuracy values with tooth form data.

x, y, and z are the 3D dimensions. L, landmark; SL, semilandmark.

Click here for additional data file.

We acknowledge the MRI platform member of the national infrastructure France-BioImaging supported by the French National Research Agency (ANR-10-INBS-04, “Investments for the future”), the labex CEMEB (ANR-10-LABX-0004) and NUMEV (ANR-10-LABX-0020) and thank Renaud Lebrun for his help with microCT imaging. We acknowledge the contribution of SFR Biosciences (UMS3444/CNRS, US8/Inserm, ENS de Lyon, UCBL) AniRa-ImmOs facility and thank Mathilde Bouchet-Combe for her help with microCT imaging. We thank Sylvie Agret for her help generating 3D surfaces.

Additional Information and Declarations

Competing Interests

Author Contributions

Animal Ethics

Data Availability

The authors declare that they have no competing interests.

Fidji Berio conceived and designed the experiments, performed the experiments, analyzed the data, prepared figures and/or tables, authored or reviewed drafts of the article, and approved the final draft.

Yann Bayle conceived and designed the experiments, performed the experiments, analyzed the data, prepared figures and/or tables, authored or reviewed drafts of the article, and approved the final draft.

Daniel Baum analyzed the data, authored or reviewed drafts of the article, and approved the final draft.

Nicolas Goudemand conceived and designed the experiments, authored or reviewed drafts of the article, and approved the final draft.

Mélanie Debiais-Thibaud conceived and designed the experiments, authored or reviewed drafts of the article, and approved the final draft.

The following information was supplied relating to ethical approvals (i.e., approving body and any reference numbers):

No approval needed (specimens from University collection or provided by marine stations). No live animals were used.

The following information was supplied regarding data availability:

The 3D surfaces are available at MorphoMuseum: Berio F., Bayle Y., Agret S., Baum D., Goudemand N., Debiais-Thibaud M., 2022. 3D models related to the publication: Hide and seek shark teeth in Random Forests: Machine learning applied to Scyliorhinus canicula. MorphoMuseuM. DOI 10.18563/journal.m3.164.

The code, landmark and semilandmark coordinates are available at GitHub: https://github.com/fberio/scanicula-teeth.

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
