# Peer review of "Hide and seek shark teeth in Random Forests: machine learning applied to Scyliorhinus canicula populations"

_PeerJ, doi:10.7717/peerj.13575_

## Round 0.1 · original submission · Major Revisions

Dear Dr. Berio

Manuscript entitled "Hide and seek shark teeth in Random Forests: Machine learning applied to Scyliorhinus canicula populations" which you submitted to PeerJ, has been reviewed. The comments of the reviewer(s) and a recommendation by the Subject Editor are included at the bottom of this letter.

The reviewer(s) and the Subject Editor have recommended major revision to your manuscript. Therefore, I invite you to respond to the reviewer(s)' comments and revise your manuscript.

Reviewer 1 has suggested that you cite specific references. You are welcome to add it/them if you believe they are relevant. However, you are not required to include these citations, and if you do not include them, this will not influence my decision.

With kind regards,
Juan Pablo Quimbayo
Academic Editor, PeerJ

Reviewer 1 ·

Basic reporting

A pdf file is provided.

Experimental design

A pdf file is provided.

Validity of the findings

A pdf file is provided.

Additional comments

This MS deals with the level of tooth shape and form differentiation between the Mediterranean and Atlantic population samples of S. canicula by using two approaches. This work is well-written and structured, and the methodology applied is considered adequate. Nevertheless, this MS needs major revision prior to acceptance based on the following issues: significant bibliographic gaps have been identified, several parts need clarification, and in several cases the assumptions should be lightened. More specifically (all comments are provided analytically in the pdf file):
 It is recommended that authors include some significant references about the species (they are provided in comments in the pdf file).
 Line 25: add the important information: "from 56 individuals".
 Lines 85-86: According to a previous study length-weight relationships of S. cannicula in the eastern Mediterranean varied significantly from those of the western and central Mediterranean.
 Lines 148-150: Authors should explain why the maturity assessment was not conducted in the current study, considering that they samples the whole specimen. This should be clarified in text.
 Line 162: the ageing methodologies used by each study should be added, considering that they are different and results may vary.
 Line 167: "Based on the von Bertalanffy growth curves" based on which exactly equations? Please clarify.
 In general, this part {167-171) needs caution considering that S. canicula vertebrae are characterized by low calcification degree and annual rings cannot be easily discriminated.
 Knowing also that in most, if not all, age-studies no age validation method has been applied, solid estimations of growth parameters cannot be provided, and thus applied in meta-analysis with high confidence. As a consequence, authors should highlight that these results need further investigation based on the above "issues".
 Lines 458-461: This part needs modification. In the case of S. canicula "Higher diet diversity was observed in males than females, in immature individuals than mature ones, regardless of sex, as well as in spring in comparison to autumn and winter. Feeding intensity seemed to be influenced mainly by sex and maturity condition." meaning that sex and season can also be considered as significant factors (sexual maturity is related positively to length).
 Lines 470-471: This statement is incorrect and needs modification. It has been found that "Both mitochondrial and nuclear microsatellite DNA data revealed a strong genetic subdivision, mainly between the western and eastern Mediterranean, whereas the Levantine Basin shared haplotypes with both areas. The geographic isolation of the Mediterranean basins seems to enforce the population genetic differentiation of the species, with the deep sea acting as a strong barrier to its dispersal."
 Lines 475-476: Is there a reference to support this statement. Usually species with pelagic larvae are influenced by hydrographic conditions. Please delete or rephrase.
 Line 477: Replace with: is probably related to the genetic......
 Lines 476-478: This part should be lightened, since the samples included in this study were not examined genetically. A sentence in the end can be added: "Nevertheless, further research examining the differentiation at both molecular and tooth morphology level of the same specimens between this regions, could confirm this assumption."

Annotated reviews are not available for download in order to protect the identity of reviewers who chose to remain anonymous.

·

Basic reporting

The article is well written, in clear and readable English.
The references are ok, and for the main scope of the article I think that the authors have provided sufficient and satisfactory background and contextualization.
The structure of the article is ok, with good figures and tables. However, for the figures that shows the morphology of teeth, as figure 3, I have one suggestion: to facilitate the reader, I suggest that the authors create two more figures. One that show the morphological comparison of the teeth of adult specimens from the Mediterranean and Atlantic populations, side by side (Atlantic male with Mediterranean male, Atlantic female with Mediterranean female), since the main morphological differences between the teeth of the two populations are found in these specimens. And a second one that shows the ontogenetic difference between neonates, juveniles and adults of each of these populations. I liked figure 3, but as a morphologist who works with fossil teeth, I think that summarizing the differences in a figure like 3 loses important visual information on the differences in the teeth of these populations which can be shown in more detail. And for the figure 5, I have one question: Does the size of the centroid have any measurement value (like centimeters or millimeters)?
The results are good, and relevant mainly because they show how RF can be a good and efficient tool for taxonomy studies that use morphometric data.

Experimental design

The article deals with new data on morphometric differences in dentition between Atlantic and Mediterranean populations of Scyliorhinus canicula, and for me the main and original point of the article was the use of RF to analyze these data. The data to be published are in agreement with the scope of the journal.
The subject addressed is well defined and the data obtained, despite being more robust when used in adult specimens of the two populations, are relevant.
There was an investigation into the differences found in the morphology of the teeth of the sampled populations of S. canicula, and it is very clear how the use of RF programs for the analysis of morphometric data (that can express these differences) was more efficient and robust than LDA. However, for the results, I felt a lack of a more complete morphological description, reporting for example the number of cusps found in hatchlings, juveniles and adults between males and females within and between the investigated populations. In my opinion, the authors were a little more concerned with showing the differences obtained between the LDA and RF analyzes used to interpret the landmarks and morphometric data of the structures, than reporting in more detail the morphological aspects of the teeth of populations analyzed.
Methodology is well explained and described.
In the abstract, I understand that stating that more than 3,000 teeth were sampled is interesting and catches the reader's attention; but I think it is more important to highlight there the number of specimens analyzed: 25 from the Mediterranean and 31 from the Atlantic. Personally I find it more relevant.

Validity of the findings

The results found are very relevant, and show mainly that morphometric data are a good source of information to be explored in structures such as teeth to help distinguish populations of the same species. Even more so if these data are added to detailed descriptions of such structures. For me, one of the highlights of the article is the demonstration of the greater efficiency of RF to analyze and interpret morphometric data (when compared to LDA) as being a great replicable methodology for studies that look for morphological differences in structures such as teeth and dermal denticles between populations of species of elasmobranchs. I see that there is great potential in using this type of analysis for morphometric data of such structures.
All underlying data are ok, however I think it is important to emphasize in the introduction that gene flow possibly occurs between the populations of S. canicula, even if at low flow, as mentioned in the final part of the discussion page 17 between lines 470 and 473, because on page 2 line 70 (introduction) the authors state there is a geographical disjunction between the populations of S. canicula, which, in my view, contradicts what was exposed in the final part of the discussion. It is clear that the analyzes of teeth sampled here for these two populations should also be carried out for other regions of the Atlantic and Mediterranean where S. canicula occurs (as the authors themselves stated) to see if there are these teeth differences between other populations. However, since there is still a genetic connection between the populations of S. canicula, the statement of a geographic disjunction can lead the reader to an interpretation of a possible speciation, which is not yet confirmed for this species.

Additional comments

Here I will expose a few corrections of small details that I found in the text:
Line 67: Eastern instead of "Western".
Line 140: I don't know if the journal allows, but usually the genus epithet is written in full at the beginning of sentences: Scyliorhinus.
Lines 445 and 465: the same as above.

---

## Round 0.2 · Major Revisions

Dear authors,

I have now received the reports from 2 referees. As you will see both reviewers found your study interesting and potentially publishable but only after a revision of the comments from reviewer 1. I agree with their comments. Thus I invite you to deeply revise your manuscript along all the lines he has suggested and to give specific attention to the methodology applied. If you feel that you can manage to take into account all the comments, please provide, together with the new manuscript, a comprehensive cover letter documenting your changes. Please note that the new manuscript will be sent again for external review.

Best regards,

Juan Pablo

Reviewer 1 ·

Basic reporting

14h March 2022
Dear Editor-in-chief,

You may find attached the revised version of the manuscript (MS) entitled “Hide and seek shark teeth in Random Forests: Machine learning applied to Scyliorhinus canicula populations”. This MS deals with the level of tooth shape and form differentiation between the Mediterranean and Atlantic population samples of S. canicula by using two approaches.

Authors have addressed several issues form previous review report. Nevertheless a major deficiency still encountered is the groupings that applied based on maturity state, as well as the calculation of growth parameters on such few measurements and by using indirect methodology.

Synoptically, the MS still needs revision mainly in terms of methodology applied: Authors should consider to delete completely the calculation of growth parameters from Methods and be specific regarding the groupings. Further minor linguistic and syntax issues are highlighted. A linguistic check is highly recommended.

A pdf file with track changes is provided for the convenience of authors.

Authors are encouraged to provide a "clean" version of the revised MS.

Sincerely,

The Reviewer

Experimental design

A pdf file is provided.

Validity of the findings

A pdf file is provided.

Additional comments

A pdf file is provided.

Annotated reviews are not available for download in order to protect the identity of reviewers who chose to remain anonymous.

·

Basic reporting

The authors responded and accepted the points made by me satisfactorily.

Experimental design

The authors responded and accepted the points made by me satisfactorily.

Validity of the findings

The authors responded and accepted the points made by me satisfactorily.

Additional comments

The authors responded and accepted the points made by me satisfactorily.

---

## Round 0.3 · Minor Revisions

Dear Dr. Berio,

The reviewer mentioned again that the part of aging estimation also need be improved to clarify some details that were included in the pdf file. Thus, my suggestion is to make a revision of these points and return the manuscript promptly to finish the review process.

All the best

Juan Pablo

Reviewer 1 ·

Basic reporting

no comment

Experimental design

pdf with comments is provided

Validity of the findings

pdf with comments is provided

Additional comments

pdf with comments is provided

Annotated reviews are not available for download in order to protect the identity of reviewers who chose to remain anonymous.

---

## Round 0.4 · accepted · Accept

Dear Dr. Berio

I am pleased to inform you that your manuscript is accepted for publication in PeerJ. Excellent job.

All the best

Juan Pablo